# Primal Wasserstein Imitation Learning

**Robert Dadashi**[*,1]**, Léonard Hussenot**[1,2]**, Matthieu Geist**[1]**, Olivier Pietquin**[1]
[1]Google Research, Brain Team
[2]Univ. de Lille, CNRS, Inria Scool, UMR 9189 CRIStAL

## Abstract

Imitation Learning (IL) methods seek to match the behavior of an agent with that of an expert. In the present work, we propose a new IL method based on a conceptually simple algorithm: Primal Wasserstein Imitation Learning (PWIL), which ties to the primal form of the Wasserstein distance between the expert and the agent state-action distributions. We present a reward function which is derived offline, as opposed to recent adversarial IL algorithms that learn a reward function through interactions with the environment, and which requires little fine-tuning. We show that we can recover expert behavior on a variety of continuous control tasks of the MuJoCo domain in a sample efficient manner in terms of agent interactions and of expert interactions with the environment. Finally, we show that the behavior of the agent we train matches the behavior of the expert with the Wasserstein distance, rather than the commonly used proxy of performance.

## 1 Introduction

Reinforcement Learning (RL) has solved a number of difficult tasks whether in games (Tesauro, 1995; Mnih et al., 2015; Silver et al., 2016) or robotics (Abbeel & Ng, 2004; Andrychowicz et al., 2020). However, RL relies on the existence of a reward function, that can be either hard to specify or too sparse to be used in practice. Imitation Learning (IL) is a paradigm that applies to these environments with *hard to specify rewards*: we seek to solve a task by learning a policy from a fixed number of demonstrations generated by an expert.

IL methods can typically be folded into two paradigms: Behavioral Cloning, or BC (Pomerleau, 1991; Bagnell et al., 2007; Ross & Bagnell, 2010) and Inverse Reinforcement Learning, or IRL (Russell, 1998; Ng et al., 2000). In BC, we seek to recover the expert's behavior by directly learning a policy that *matches* the expert behavior in some sense. In IRL, we assume that the demonstrations come from an agent that acts optimally with respect to an unknown reward function that we seek to recover, to subsequently train an agent on it. Although IRL methods introduce an intermediary problem (i.e. recovering the environment's reward) they are less sensitive to distributional shift (Pomerleau, 1991), they generalize to environments with different dynamics (Piot et al., 2013), and they can recover a near-optimal agent from suboptimal demonstrations (Brown et al., 2019; Jacq et al., 2019).

However, IRL methods are usually based on an iterative process alternating between reward estimation and RL, which might result in poor sample-efficiency. Earlier IRL methods (Ng et al., 2000; Abbeel & Ng, 2004; Ziebart et al., 2008) require multiple calls to a Markov decision process solver (Puterman, 2014), whereas recent adversarial IL approaches (Finn et al., 2016; Ho & Ermon, 2016; Fu et al., 2018) interleave the learning of the reward function with the learning process of the agent. Adversarial IL methods are based on an adversarial training paradigm similar to Generative Adversarial Networks (GANs) (Goodfellow et al., 2014), where the learned reward function can be thought of as the confusion of a discriminator that learns to differentiate expert transitions from non expert ones. These methods are well suited to the IL problem since they implicitly minimize an $f$-divergence between the state-action distribution of an expert and the state-action distribution of the learning agent (Ghasemipour et al., 2019; Ke et al., 2019). However the interaction between a generator (the policy) and the discriminator (the reward function) makes it a minmax optimization problem, and therefore comes with practical challenges that might include training instability, sensitivity to hyperparameters and poor sample efficiency.

---

[*]Correspondence to Robert Dadashi: `dadashi@google.com`.

In this work, we use the Wasserstein distance as a measure between the state-action distributions of the expert and of the agent. Contrary to $f$-divergences, the Wasserstein distance is a true distance, it is smooth and it is based on the geometry of the metric space it operates on. The Wasserstein distance has gained popularity in GAN approaches (Arjovsky et al., 2017) through its dual formulation which comes with challenges (see Section 5). Our approach is novel in the fact that we consider the problem of minimizing the Wasserstein distance through its primal formulation. Crucially, the primal formulation prevents the minmax optimization problem, and requires little fine tuning.

We introduce a reward function computed offline based on an upper bound of the primal form of the Wasserstein distance. As the Wasserstein distance requires a distance between state-action pairs, we show that it can be hand-defined for locomotion tasks, and that it can be learned from pixels for a hand manipulation task. The inferred reward function is non-stationary, like adversarial IL methods, but it is not re-evaluated as the agent interacts with the environment, therefore the reward function we define is computed *offline*. We present a true distance to compare the behavior of the expert and the behavior of the agent, rather than using the common proxy of performance with respect to the true return of the task we consider (as it is unknown in general). Our method recovers expert behaviour comparably to existing state-of-the-art methods while being based on significantly fewer hyperparameters; it operates even in the extreme low data regime of demonstrations, and is the first method that makes *Humanoid* run with a single (subsampled) demonstration.

## 2 BACKGROUND AND NOTATIONS

**Markov decision processes.** We describe environments as episodic Markov Decision Processes (MDP) with finite time horizon (Sutton & Barto, 2018) $(\mathcal{S}, \mathcal{A}, \mathcal{P}, r, \gamma, \rho_0, T)$, where $\mathcal{S}$ is the state space, $\mathcal{A}$ is the action space, $\mathcal{P}$ is the transition kernel, $r$ is the reward function, $\gamma$ is the discount factor, $\rho_0$ is the initial state distribution and $T$ is the time horizon. We will denote the dimensionality of $\mathcal{S}$ and $\mathcal{A}$ as $|\mathcal{S}|$ and $|\mathcal{A}|$ respectively. A policy $\pi$ is a mapping from states to distributions over actions; we denote the space of all policies by $\Pi$. In RL, the goal is to learn a policy $\pi^*$ that maximizes the expected sum of discounted rewards it encounters, that is, the expected return. Depending on the context, we might use the concept of a cost $c$ rather than a reward $r$ (Puterman, 2014), which essentially moves the goal of the policy from maximizing its return to minimizing its cumlative cost.

**State action distributions.** Suppose a policy $\pi$ visits the successive states and actions $s_1, a_1, \ldots, s_T, a_T$ during an episode, we define the empirical state-action distribution $\hat{\rho}_\pi$ as: $\hat{\rho}_\pi = \frac{1}{T} \sum_{t=1}^{T} \delta_{s_t, a_t}$, where $\delta_{s_t, a_t}$ is a Dirac distribution centered on $(s_t, a_t)$. Similarly, suppose we have a set of expert demonstrations $\mathcal{D} = \{s^e, a^e\}$ of size $D$, then the associated empirical expert state-action distribution $\hat{\rho}_e$ is defined as: $\hat{\rho}_e = \frac{1}{D} \sum_{(s,a) \in \mathcal{D}} \delta_{s,a}$.

**Wasserstein distance.** Suppose we have the metric space $(M, d)$ where $M$ is a set and $d$ is a metric on $M$. Suppose we have $\mu$ and $\nu$ two distributions on $M$ with finite moments, the $p$-th order Wasserstein distance (Villani, 2008) is defined as $\mathcal{W}_p^p(\mu, \nu) = \inf_{\theta \in \Theta(\mu, \nu)} \int_{M \times M} d(x, y)^p d\theta(x, y)$, where $\Theta(\mu, \nu)$ is the set of all couplings between $\mu$ and $\nu$. In the remainder, we only consider distributions with finite support. A coupling between two distributions of support cardinal $T$ and $D$ is a doubly stochastic matrix of size $T \times D$. We note $\Theta$ the set of all doubly stochastic matrices of size $T \times D$:

$$\Theta = \left\{ \theta \in \mathbb{R}_+^{T \times D} \mid \forall j \in [1 : D], \sum_{i'=1}^{T} \theta[i', j] = \frac{1}{D}, \forall i \in [1 : T], \sum_{j'=1}^{D} \theta[i, j'] = \frac{1}{T} \right\}.$$

The Wasserstein distance between distributions of state-action pairs requires the definition of a metric $d$ in the space $(\mathcal{S}, \mathcal{A})$. Defining a metric in an MDP is non trivial (Ferns et al., 2004; Mahadevan & Maggioni, 2007); we show an example where the metric is learned from demonstrations in Section 4.4. For now, we assume the existence of a metric $d : (\mathcal{S}, \mathcal{A}) \times (\mathcal{S}, \mathcal{A}) \mapsto \mathbb{R}^+$.

## 3 METHOD

We present the theoretical motivation of our approach: the minimization of the Wasserstein distance between the state-action distributions of the agent and the expert. We introduce a reward based on an upper-bound of the primal form of the Wasserstein distance inferred from a relaxation of

the optimal coupling condition, and present the resulting algorithm: Primal Wasserstein Imitation Learning (PWIL).

## 3.1 Wasserstein distance minimization

Central to our approach is the minimization of the Wasserstein distance between the state-action distribution of the policy we seek to train $\hat{\rho}_\pi$ and the state-action distribution of the expert $\hat{\rho}_e$. In other words, we aim at optimizing the following problem:

$$\inf_{\pi \in \Pi} \mathcal{W}_p^p(\hat{\rho}_\pi, \hat{\rho}_e) = \inf_{\pi \in \Pi} \inf_{\theta \in \Theta} \sum_{i=1}^{T} \sum_{j=1}^{D} d((s_i^\pi, a_i^\pi), (s_j^e, a_j^e))^p \theta[i, j]. \tag{1}$$

In the rest of the paper, we only consider the 1-Wasserstein ($p = 1$ in Equation 1) and leave the extensive study of the influence of the order $p$ for future work. We can interpret the Wasserstein distance using the earth's movers analogy (Villani, 2008). Consider that the state-action pairs of the expert are $D$ *holes* of mass $D^{-1}$ and that the state-action pairs of the policy are *piles of dirt* of mass $T^{-1}$. A coupling $\theta$ is a transport strategy between the piles of dirt and the holes, where $\theta[i, j]$ stands for how much of the pile of dirt $i$ should be moved towards the hole $j$. The optimal coupling is the one that minimizes the distance that the earth mover travels to put all piles of dirt to holes. Note that to compute the optimal coupling, we need knowledge of the locations of all piles of dirt. In the context of RL, this means having access to the full trajectory generated by $\pi$. From now on, we write $\theta_\pi^*$ as the optimal coupling for the policy $\pi$, that we inject in Equation (1):

$$\theta_\pi^* = \arg\min_{\theta \in \Theta} \sum_{i=1}^{T} \sum_{j=1}^{D} d((s_i^\pi, a_i^\pi), (s_j^e, a_j^e)) \theta[i, j]$$

$$\inf_{\pi \in \Pi} \mathcal{W}_1(\hat{\rho}_\pi, \hat{\rho}_e) = \inf_{\pi \in \Pi} \sum_{i=1}^{T} c_{i,\pi}^* \text{ with } c_{i,\pi}^* = \sum_{j=1}^{D} d((s_i^\pi, a_i^\pi), (s_j^e, a_j^e)) \theta_\pi^*[i, j]. \tag{2}$$

In Equation (2), we have introduced $c_{i,\pi}^*$, which we interpret as a cost to minimize using RL. As $c_{i,\pi}^*$ depends on the optimal coupling $\theta_\pi^*$, we can only define $c_{i,\pi}^*$ at the very end of an episode. This can be problematic if an agent learns in an online manner or in large time-horizon tasks. Thus, we introduce an upper bound to the Wasserstein distance that yields a cost we can compute online, based on a suboptimal coupling strategy.

## 3.2 Greedy coupling

In this section we introduce the greedy coupling $\theta_\pi^g \in \Theta$, defined recursively for $1 \leq i \leq T$ as:

$$\theta_\pi^g[i, :] = \arg\min_{\theta[i,:] \in \Theta_i} \sum_{j=1}^{D} d((s_i^\pi, a_i^\pi), (s_j^e, a_j^e)) \theta[i, j] \tag{3}$$

$$\text{with } \Theta_i = \left\{ \theta[i, :] \in \mathbb{R}_+^D \,\Big|\, \underbrace{\sum_{j'=1}^{D} \theta[i, j'] = \frac{1}{T}}_{\text{constraint}(a)}, \underbrace{\forall k \in [1 : D], \sum_{i'=1}^{i-1} \theta_g[i', k] + \theta[i, k] \leq \frac{1}{D}}_{\text{constraint}(b)} \right\}.$$

Similarly to Equation (1), Equation (3) can be interpreted using the earth mover's analogy. Contrary to Equation (1) where we assume knowledge of the positions of the $T$ piles of dirts, we now consider that they appear sequentially, and that the earth's mover needs to transport the new pile of dirt to holes *right when it appears*. To do so, we derive the distances to all holes and move the new pile of dirt to the closest remaining available holes, hence the *greedy* nature of it. In Equation (3), the constraint ($a$) means that all the dirt that appears at the $i$-th timestep needs to be moved, and the constraint ($b$) means that we cannot fill the holes more than their capacity $\frac{1}{D}$. We show the difference between the greedy coupling and the optimal coupling in Figure 1, and provide the pseudo-code to derive it in Algorithm 1.

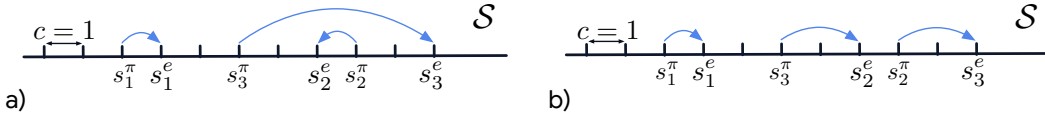

a)                                                                                b)

Figure 1: Illustration of the difference between the a) greedy coupling and the b) optimal coupling. We present an MDP where we drop the dependency on the action. The state space is $\mathbb{R}$ and the metric associated is the Euclidean distance. We note the states visited by the policy: $s_1^\pi, s_2^\pi, s_3^\pi$ and the states visited by the expert: $s_1^e, s_2^e, s_3^e$. When the policy encounters the state $s_2^\pi$, and because we do not know $s_3^\pi$ yet, the greedy coupling strategy consists in coupling it with $s_2^e$ although the optimal coupling strategy would be to couple it with $s_3^e$. Note that the total cost (omitting the constant coupling multiplication factor) with the greedy coupling is 7 whereas the total cost with the optimal coupling is 5 . This highlights that the optimal coupling needs knowledge about the whole policy's trajectory to be derived.

We can now define an upper bound of the Wasserstein distance using the greedy coupling (since by definition it is suboptimal):

$$\inf_{\pi \in \Pi} \mathcal{W}_1(\hat{\rho}_\pi, \hat{\rho}_e) = \inf_{\pi \in \Pi} \sum_{i=1}^{T} \sum_{j=1}^{D} d((s_i^\pi, a_i^\pi), (s_j^e, a_j^e)) \theta_\pi^*[i,j]$$

$$\leq \inf_{\pi \in \Pi} \sum_{i=1}^{T} \sum_{j=1}^{D} d((s_i^\pi, a_i^\pi), (s_j^e, a_j^e)) \theta_\pi^g[i,j]. \tag{4}$$

In Section 4, we empirically validate this bound. We infer a cost from Equation (1) at each timestep $i$:

$$c_{i,\pi}^g = \sum_{j=1}^{D} d((s_i^\pi, a_i^\pi), (s_j^e, a_j^e)) \theta_\pi^g[i,j]. \tag{5}$$

Note that the greedy coupling $\theta_\pi^g[t,.]$ defined in Equation (3) depends on all the previous state-actions visited by the policy $\pi$, which makes the cost $c_{i,\pi}^g$ non-stationary, similarly to adversarial IL methods. We can infer a reward from the cost, $r_{i,\pi} = f(c_{i,\pi}^g)$ where $f$ is a monotonically decreasing function. This reward function is an episodic history dependent reward function $r(s_0, a_0, \ldots, s_t, a_t)$ where $r$ is uniquely defined from expert demonstrations (hence is said to be derived *offline*). Crucially, we have defined a reward function that we seek to maximize, without introducing an inner minimization problem (which is the case with adversarial IL approaches).

We can now derive an algorithm building on this reward function (Algorithm 1). The algorithm presented is written using pseudo-code for a generic agent $A$ which implements a policy $\pi_A$, it observes tuples $(s, a, r, s')$ and possibly updates its components. The runtime of the algorithm for a single reward step computation has complexity $\mathcal{O}((|\mathcal{S}| + |\mathcal{A}|)D + \frac{D^2}{T})$ (see Appendix E.2 for details), and therefore has complexity $\mathcal{O}((|\mathcal{S}| + |\mathcal{A}|)DT + D^2)$ for computing rewards across the entire episode. Note that in the case where $T = D$, computing the greedy coupling is simply a lookup of the expert state-action pair that minimizes the distance with the agent state-action pair, followed by a pop-out of this minimizing expert state-action pair from the set of expert demonstrations.

Algorithm 1 introduces an agent with the capability to *observe* and *update* itself, which is directly inspired from ACME's formalism of agents (Hoffman et al., 2020), and is general enough to characterize a wide range of agents. We give an example rundown of the algorithm in Section E.1.

## 4   EXPERIMENTS

In this section, we present the implementation of PWIL and perform an empirical evaluation, based on the ACME framework (Hoffman et al., 2020). We test PWIL on MuJoCo locomotion tasks and compare it to the state-of-the-art GAIL-like algorithm DAC (Kostrikov et al., 2019) and against the common BC baseline. As DAC is based on TD3 (Fujimoto et al., 2018) which is a variant of DDPG (Lillicrap et al., 2016), we use a DDPG-based agent for fair comparison: D4PG (Barth-Maron et al., 2018). We also provide a proof-of-concept experiment on a door opening task where the demonstrations do not include actions and are pixel-based and therefore the metric of the

---

**Algorithm 1** PWIL: Primal Wasserstein Imitation Learning

---

**Input:** Expert demonstrations $\mathcal{D} = \{s_j^e, a_j^e\}_{j \in [1:D]}$, Agent $A$, number of episodes $N$
**for** $k = 1$ **to** $N$ **do**
    Copy expert demonstrations with weight: $\mathcal{D}' := \{s_j^e, a_j^e, w_j^e\}_{j \in [1:D]}$, with $w_j^e = \frac{1}{D}$
    Reset environment, initial state $s$
    **for** $i = 1$ **to** $T$ **do**
        Take action $a := \pi_A(s)$, observe next state $s'$
        Initialize weight $w^\pi := \frac{1}{T}$, cost $c := 0$
        **while** $w^\pi > 0$ **do**
            Compute $s^e, a^e, w^e := \arg \min_{(s^e, a^e, w^e) \in \mathcal{D}'} d((s, a), (s^e, a^e))$
            **if** $w^\pi \geq w^e$ **then**
                $c := c + w^e d((s, a), (s^e, a^e))$
                $w^\pi := w^\pi - w^e$
                $\mathcal{D}'.\text{pop}(s^e, a^e, w^e)$
            **else**
                $c := c + w^\pi d((s, a), (s^e, a^e))$
                $w^e := w^e - w^\pi$
                $w^\pi := 0$
        $r := f(c)$
        $A$ observes $(s, a, r, s')$, updates itself

---

MDP has to be learned. In this setup, we use SAC as the direct RL method. We answer the following questions: 1) Does PWIL recover expert behavior? 2) How sample efficient is PWIL? 3) Does PWIL actually minimize the Wasserstein distance between the distributions of the agent and the expert? 4) Does PWIL extend to visual based observations where the definition of an MDP metric is not straightforward? We also show the dependence of PWIL on multiple of its components through an ablation study. The complete description of the experiments is given in Appendix A. We provide experimental code and videos of the trained agents here: `https://sites.google.com/view/wasserstein-imitation`.

## 4.1 IMPLEMENTATION

We test PWIL following the same evaluation protocol as GAIL and DAC, on six environments from the OpenAI Gym MuJoCo suite (Todorov et al., 2012; Brockman et al., 2016): Reacher-v2, Ant-v2, Humanoid-v2, Walker2d-v2, HalfCheetah-v2 and Hopper-v2. For each environment, multiple demonstrations are gathered using a D4PG agent trained on the actual reward of these environments (which was chosen as it is a base agent in the ACME framework). We subsample demonstrations by a factor of 20: we select one out of every 20 transitions with a random offset at the beginning of the episode (expect for Reacher for which we do not subsample since the episode length is 50). The point of subsampling is to simulate a low data regime, so that a pure behavioral cloning approach suffers the behavioral drift and hence does not perform to the level of the expert. We run experiments with multiple number of demonstrations: $\{1, 4, 11\}$ (consistently with the evaluation protocol of DAC).

Central to our method is a metric between state-action pairs. We use the *standardized Euclidean distance* which is the L2 distance on the concatenation of the observation and the action, weighted along each dimension by the inverse standard deviation of the expert demonstrations. From the cost $c_i$ defined in Equation (5), we define the following reward:

$$r_i = \alpha \exp(-\frac{\beta T}{\sqrt{|\mathcal{S}| + |\mathcal{A}|}} c_i). \tag{6}$$

For all environments, we use $\alpha = 5$ and $\beta = 5$. The scaling factor $\frac{T}{\sqrt{|\mathcal{S}|+|\mathcal{A}|}}$ acts as a normalizer on the dimensionality of the state and action spaces and on the time horizon of the task. We pick $f : x \mapsto \exp(-x)$ as the monotonically decreasing function.

We test our method against a state-of-the-art imitation learning algorithm: DAC and BC. DAC is based on GAIL: it discriminates between expert and non-expert states and it uses an off-policy algorithm TD3 which is a variant of DDPG instead of the on-policy algorithm TRPO (Schulman

et al., 2015). We used the open-source code provided by the authors. The Humanoid environment is not studied in the original DAC paper. The hyperparameters the authors provided having poor performance, we led an exhaustive search detailed in Section A.2, and present results for the best hyperparameters found.

For our method, we use a D4PG agent (see details in Appendix A) and provide additional experiments with different direct RL agents in Appendix C. We initialize the replay buffer by filling it with expert state-action pairs with maximum reward $\alpha$ (however, note that we used subsampled trajectories which makes it an approximation to replaying the actual trajectories). We train PWIL and DAC in the limit of 1M environment interactions (2.5M for Humanoid) on 10 seeds. Every 10k environment steps, we perform 10-episode rollouts per seed of the policy without exploration noise and report performance with respect to the environment's original reward in Figure 2.

## 4.2 RESULTS

In Figure 2, PWIL shows improvement on final performance for Hopper, Ant, HalfCheetah and Humanoid over DAC, similar performance for Walker2d, and is worse on Reacher. On Humanoid, although we ran an exhaustive hyperparameter search (see Section A.2) DAC has poor performance, thus exemplifying the brittleness of GAIL-like approaches. On the contrary, PWIL has near-optimal performance, even with a single demonstration.

In terms of sample efficiency, DAC outperforms PWIL on HalfCheetah, Ant and Reacher and has similar performance for Hopper and Walker2d. Note that ablated versions of PWIL are more sample-efficient than DAC on HalfCheetah and Ant (Section B). This seems rather contradictory with the claim that GAIL-like methods have poor sample efficiency. However the reward function defined in DAC requires careful tuning (e.g. architecture, optimizer, regularizers, learning rate schedule) which demands experimental cycles and therefore comes with a large number of environment interactions.

Remarkably, PWIL is the first method able to consistently learn policies on Humanoid with an average score over 7000 even with a single demonstration, which means that the agent can actually *run*, (other works consider Humanoid is "solved" for a score of 5000 wich corresponds to *standing*).

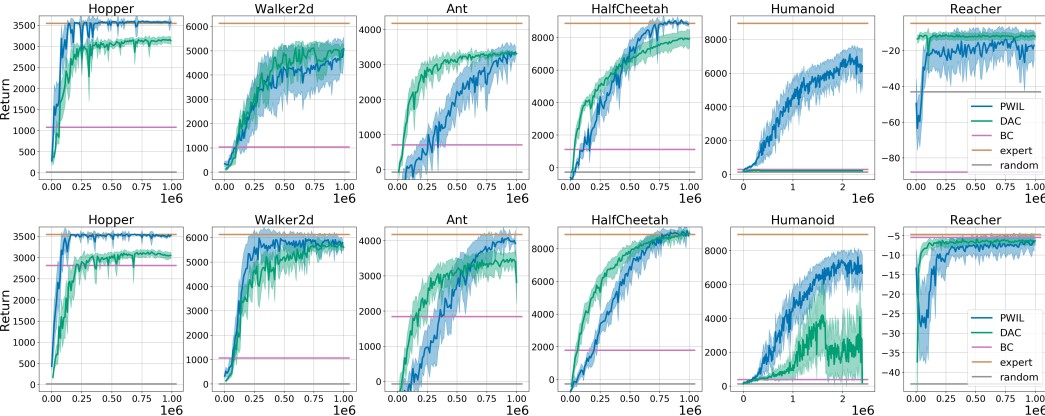

Figure 2: Mean and standard deviation return of the evaluation policy over 10 rollouts and 10 seeds, reported every 10k environment steps. The return is in term of the environment's original reward. Top row: 1 demonstration, bottom row: 11 demonstrations.

**Wasserstein distance.** In Figure 3, we show the Wasserstein distance to expert demonstrations throughout the learning procedure for PWIL and DAC. The Wasserstein distance is evaluated at the end of an episode using a transportation simplex (Bonneel et al., 2011); we used the implementation from Flamary & Courty (2017). For both methods, we notice that the distance decreases while learning. PWIL leads to a smaller Wasserstein distance than DAC on all environments but HalfCheetah where it is similar and Reacher where it is larger. This should not come as a surprise since PWIL's objective is to minimize the Wasserstein distance. PWIL defines a reward from an upper bound to the Wasserstein distance between the state-action distributions of the expert and the agent, that we include as well in Figure 3. Notice that our upper bound is "empirically tight", which validates the choice of the greedy

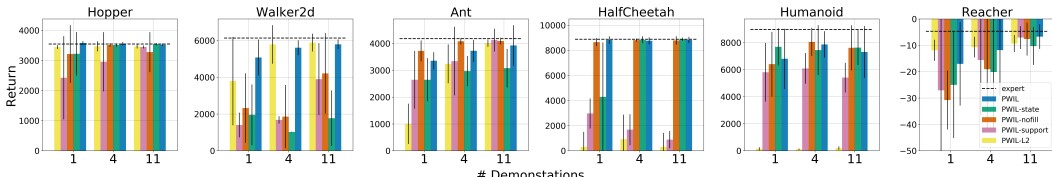

Figure 3: Mean of the Wasserstein distance between the state-action distribution of the evaluation policy and the state-action distribution of the expert over 10 rollouts and 10 seeds. Agents were trained with 11 demonstrations. For PWIL, we include the upper bound on the Wasserstein distance based on the greedy coupling defined in Equation (4).

coupling as an approximation to the optimal coupling. We emphasize that this distance is useful in real settings of IL, regardless of the method used, where we cannot have access to the actual reward of the task.

## 4.3 ABLATION STUDY

In this section, we present the evaluation performance of PWIL in the presence of ablations and report final performances in Figure 4. The learning curves for all ablations can be found in Appendix B. We keep the hyperparameters of the original PWIL agent fixed.

Figure 4: Mean and standard deviation of the evaluation performance of PWIL and variants at the 1M environment interactions mark (2.5M for Humanoid). Results are computed over 10 seeds and 10 episodes for each seed.

**PWIL-state.** In this version of PWIL, we do not assume that we have access to the actions taken by the expert. Therefore, we try to match the state distribution of the agent with the state distribution of the expert (instead of the state-action distribution). The setup is referred to as Learning from Observation (LfO) (Torabi et al., 2018; Edwards et al., 2019). The reward is defined similarly to PWIL, using a state distance rather than a state-action distance. Note that in this version, we cannot pre-fill the replay buffer with expert state-action pairs since we do not have access to actions. Remarkably, PWIL-state recovers non-trivial behaviors on all environments.

**PWIL-L2.** In this version of PWIL, we use the Euclidean distance between state-action pairs, rather than the standardized Euclidean distance. In other words, we do not weight the state-action distance by the inverse standard deviation of the expert demonstrations along each dimension. There is a significant drop in performance for all environments but Hopper-v2. This shows that the performance of PWIL is sensitive to the quality of the MDP metric.

**PWIL-nofill.** In this version, we do not prefill the replay buffer with expert transitions. This leads to a drop in performance which is significant for Walker and Ant. This should not come as a surprise since a number of IL methods leverage the idea of expert transitions in the replay buffer (Reddy et al., 2020; Hester et al., 2018).

**PWIL-support.** In this version of PWIL, we define the reward as a function of the following cost:

$$\forall i \in [1:T], \; c_{i,\pi} = \inf_{\substack{\theta_1,\dots,\theta_D \in \mathbb{R} \\ \sum_{j=1}^{D} \theta_j \le \frac{1}{T}}} \sum_{j=1}^{D} d((s_i^\pi, a_i^\pi), (s_j^e, a_j^e))\theta_j.$$

We can interpret this cost in the context of Section 3 as the problem of moving piles of dirt of mass $\frac{1}{T}$ into holes of infinite capacity. It is thus reminiscent of methods that consider IL as a support

estimation problem (Wang et al., 2019; Brantley et al., 2020). This leads to a significant drop in performance for Ant, Walker and HalfCheetah.

## 4.4 DOOR OPENING TASK: REWARD FROM PIXEL-BASED OBSERVATIONS

In this section, we show that PWIL extends to visual-based setups, where the MDP metric has to be learned. We use the door opening task from Rajeswaran et al. (2018), with *human* demonstrations generated with Haptix (Kumar & Todorov, 2015). The goal of the task is for the controlled hand to open the door (Figure 5) whose location changes at each episode. We add an early termination constraint when the door is opened (hence suppressing the incentive for survival). We assume that we can only access the high-dimensional visual rendering of the demonstrations rather than the internal state (and hence have no access to actions, that is LfO setting).

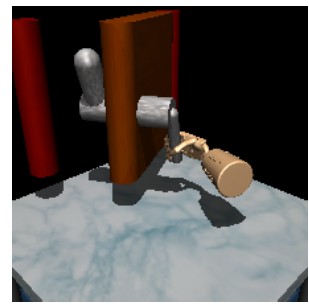

Figure 5: Visual rendering of the door opening task.

The PWIL reward is therefore defined using a distance between the visual rendering of the current state and the visual rendering of the demonstrations. This distance is learned offline through self-supervised learning: we construct an embedding of the frames using Temporal Cycle-Consistency Learning (TCC) (Dwibedi et al., 2019); TCC builds an embedding by aligning frames of videos of a task coming from multiple examples (in this case the demonstrations). The distance learned is thus the L2 distance between embeddings. The distance is learned on the 25 human demonstrations provided by Rajeswaran et al. (2018) which are then re-used to define the PWIL reward. From the cost $c_i$ defined in Equation (5), we define the following reward: $r_i = \alpha \exp(-\beta T c_i)$, with $\alpha = 5, \beta = 1$. We used SAC (Haarnoja et al., 2018a) as the forward RL algorithm, which was more robust than D4PG on the task. We train the agent on 1.5M environment steps and show that we indeed recover expert behaviour and consistently solve the task in Figure 6.

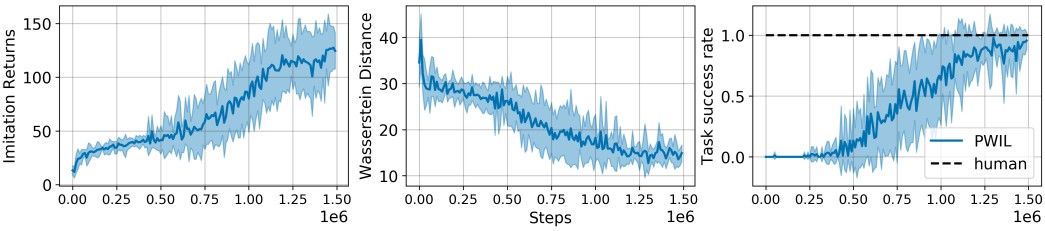

Figure 6: We present the average and standard deviation of evaluation performance of PWIL on the door environment. Metrics are computed every 10k environment steps and aggregated over 10 seeds and 10 rollouts. Left: the imitation returns of the agent. Center: Wasserstein distance in the embedding space to expert demonstrations. Right: Success rate of the agent, *i.e.* ratio of episodes where the agent manages to open the door.

## 5 RELATED WORK

**Adversarial IL.** Similarly to our method, adversarial IL methods aim at matching the state-action distribution of the agent with the distribution of the expert, using different measures of similarity. For instance, GAIL (Ho & Ermon, 2016) considers the Shannon-Jensen divergence while AIRL (Fu et al., 2018) considers the Kullback-Leibler divergence. In the context of GANs, Arjovsky et al. (2017) show that replacing the $f$-divergence by the Wasserstein distance through the Kantorovich-Rubinstein duality (Villani, 2008) leads to better training stability, which a number of methods in IL have leveraged (Li et al., 2017; Lacotte et al., 2019; Kostrikov et al., 2019; Xiao et al., 2019; Liu et al., 2020; Pacchiano et al., 2020). However, the Kantorovich-Rubinstein duality only holds for the $\mathcal{W}_1$ distance (Peyré et al., 2019), its implementation comes with a number of practical approximations (*e.g.*, weight clipping to ensure Lipchitz continuity). Although these methods are based on the Wasserstein distance through its dual formulation, they drastically differ from ours as they rely on a minmax optimization problem. DAC (Kostrikov et al., 2019) demonstrates that adversarial IL

approaches, while based on GAIL, can be sample efficient. However, they rely on a carefully tuned discriminator (e.g. network architecture, regularization strategy, scheduled learning rates) which requires to interact with the environment. In contrast, the reward function we present relies on only two hyperparameters.

**Expert support estimation.** Another line of research in IL consists in estimating the state-action support of the expert and define a reward that encourages the agent to stay on the support of the expert (Piot et al., 2014; Schroecker & Isbell, 2017; Wang et al., 2019; Brantley et al., 2020; Reddy et al., 2020). Note that the agent might stay on the support of the expert without recovering its state-action distribution. Soft Q Imitation Learning (Reddy et al., 2020) assigns a reward of 1 to all expert transitions and 0 to all transitions the expert encounters; the method learns to recover expert behavior by balancing out the expert transitions with the agent transitions in the replay buffer of a value-based agent. Random Expert Distillation (Wang et al., 2019) estimates the support by using a neural network trained on expert transitions whose target is a fixed random neural network. Disagreement Regularized Imitation Learning (Brantley et al., 2020) estimates the expert support through the variance of an ensemble of BC agents, and use a reward based on the distance to the expert support and the KL divergence with the BC policies. PWIL differs from these methods since it is based on the distance to the support of the expert, with a support that *shrinks* through "pop-outs" to enforce the agent to visit the whole support. We showed in the ablation study that "pop-outs" are *sine qua non* for recovering expert behaviour (see *PWIL-support*).

**Trajectory-Based IL.** Similarly to the door opening experiment, Aytar et al. (2018) learn an embedding offline and reinforces a reward based on the distance in the embedding space to a single demonstration. However, it is well suited for deterministic environments since they use a single trajectory that the agent aims at reproducing while the door experiment shows that PWIL is able to generalize as the door and handle locations are picked at random. Peng et al. (2018) reinforces a temporal reward function (the reward at time $t$ is based on the distance of the state of the agent to the state of the expert at time $t$). Contrary to PWIL, the imitation reward function is added to a task-dependent reward function; the distance to expert demonstrations is defined along handpicked dimensions of the observation space; the temporal constraint makes it hard to generalize to environment with stochastic initial states.

**Offline reward estimation.** Existing approaches derive a reward function without interaction with the environment (Boularias et al., 2011; Klein et al., 2013; 2012; Piot et al., 2016). They typically require strong assumptions on the structure of the reward (*e.g.* linearity in some predefined features). PWIL also derives a reward offline, without structural assumptions, however it is non-stationary since it depends on the state-action visited in the episode.

**MMD-IL.** Generative Moment Matching Imitation Learning (Kim & Park, 2018) assumes a metric on the MDP and aims at minimizing the Maximum Mean Discrepancy (MMD) between the agent and the expert. However, the MMD (like the Wasserstein distance) requires complete rollouts of the agent, and does not present a natural relaxation like the greedy coupling. GMMIL is based on an on-policy agent which makes it significantly less sample-efficient than PWIL. We were not able to implement a version of MMD based on an off-policy agent.

## 6 Conclusion

In this work, we present Imitation Learning as a distribution matching problem and introduce a reward function which is based on an upper bound of the Wasserstein distance between the state-action distributions of the agent and the expert. A number of IL methods are developed on synthetic tasks, where the evaluation of the IL method can be done with the actual return of the task. We emphasize that in our work, we present a direct measure of similarity between the expert and the agent, that we can thus use in real IL settings, that is in settings where the reward of the task cannot be specified.

The reward function we introduce is learned offline, in the sense that it does not need to be updated with interactions with the environment. It requires little tuning (2 hyperparameters) and it can recover near expert performance with as little as 1 demonstration on all considered environments, even the challenging Humanoid. Finally our method extends to the harder visual-based setting, based on a metric learned offline from expert demonstrations using self-supervised learning.

ACKNOWLEDGMENTS

We thank Zafarali Ahmed, Adrien Ali Taiga, Gabriel Dulac-Arnold, Johan Ferret, Alexis Jacq and Saurabh Kumar for their feedback on an earlier version of the manuscript. We thank Debidatta Dwibedi and Ilya Kostrikov for helpful discussions.

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

# A  LOCOMOTION EXPERIMENTS

## A.1  PWIL IMPLEMENTATION

The agent we use is D4PG (Barth-Maron et al., 2018). We use the default architecture from the ACME framework (Hoffman et al., 2020).

The actor architecture is a 4-layer neural network: the first layer has size 256 with tanh activation and layer normalization (Ba et al., 2016), the second layer and third layer have size 256 with elu activation (Clevert et al., 2016), the last layer is of size the dimension of the action space, with a tanh activation scaled to the action range of the environment. To enable sufficient exploration with use a Gaussian noise layer on top of the last layer with standard deviation $\sigma = 0.2$, that we clip to the action range of the environment. We evaluate the agent without exploration noise.

For the critic network we use a 4-layer neural network: the first layer has size 512 with tanh activation and layer normalization, the second layer is of size 512 with elu activation, the third layer is of size 256 with elu activation, the last layer is of dimension 201 with a softmax activation. The 201 neurons stand for the weights of the distribution supported within equal distance in the range $[-150, 150]$, as a categorical distribution (Bellemare et al., 2017).

We use the Adam optimizer (Kingma & Ba, 2015) with $\lambda_a = 5 \times 10^{-5}$ for the actor and $\lambda_c = 7 \times 10^{-5}$ for the critic. We use a batch size of 256. We clip both gradients from the critic and the actor to limit their L2 norm to $40$.

We use a replay buffer of size $10^6$, a discount factor $\theta = 0.99$ and $n$ step returns with $n = 5$. We prefill the replay buffer with 50000 state-action pairs from the set of demonstrations (which means that we put multiple times the same expert transitions in the buffer). We perform updates on the actor and the critic every $k = 4$ interactions with the environment. The hyperparameters search is detailed in Table 1.

| Parameters | Values |
|---|---|
| $\lambda_a$ | $10^{-5}, 5 \times 10^{-5}, 7 \times 10^{-5}, 10^{-4}$ |
| $\lambda_c$ | $10^{-5}, 5 \times 10^{-5}, 7 \times 10^{-5}, 10^{-4}$ |
| $\sigma$ | $0.1, 0.2, 0.3$ |
| $k$ | $2, 4, 8, 16$ |

Table 1: DDPG hyperparameters search.

For the reward function, we did run a hyperparameter search on $\alpha$ and $\beta$ with the following values for $\alpha$ and $\beta$ in $\{1, 5, 10\}$.

## A.2  DAC IMPLEMENTATION

We used the open-source implementation of DAC provided by Kostrikov et al. (2019). For Humanoid (which was not reported in the paper), we did run a large hyperparameter search described in Table 2. Out of the 729 experiments, 7 led to an average performance above 1000 at the 2.5M training environment steps mark.

| Actor learning rate (lr) | Critic lr | Discriminator lr | WGAN regularizer | Exploration noise | Decay lr |
|---|---|---|---|---|---|
| **0.001**, 0.0005, 0.0001 | 0.001, **0.0005**, 0.0001 | **0.001**, 0.0005, 0.0001 | 5, **10**, 20 | **0.1**, 0.2, 0.3 | 0.3, 0.5, **0.8** |

Table 2: Hyperparameter search on Humanoid for 11 demonstrations. (tested/**best**)

## A.3  BC IMPLEMENTATION

We used a 3-layer neural network, the first layer has size 128 with relu activation, the second layer has size 64 and the last layer has size of the action space with a tanh activation scaled to the action range of the environment. We found out that normalizing the observations with the average and standard deviation of the expert demonstrations' observations helped performance. We trained the network using the mean squared error as a loss, with an Adam optimizer. We ran an hyperpameter search on the learning rate in $\{10^{-5}, 10^{-4}, 10^{-3}\}$ and on the batch size $\{128, 256\}$.

## A.4 PWIL LEARNING CURVES

We present the learning curves of PWIL, in terms of the reward defined in Equation 6 as well as the original reward of the task.

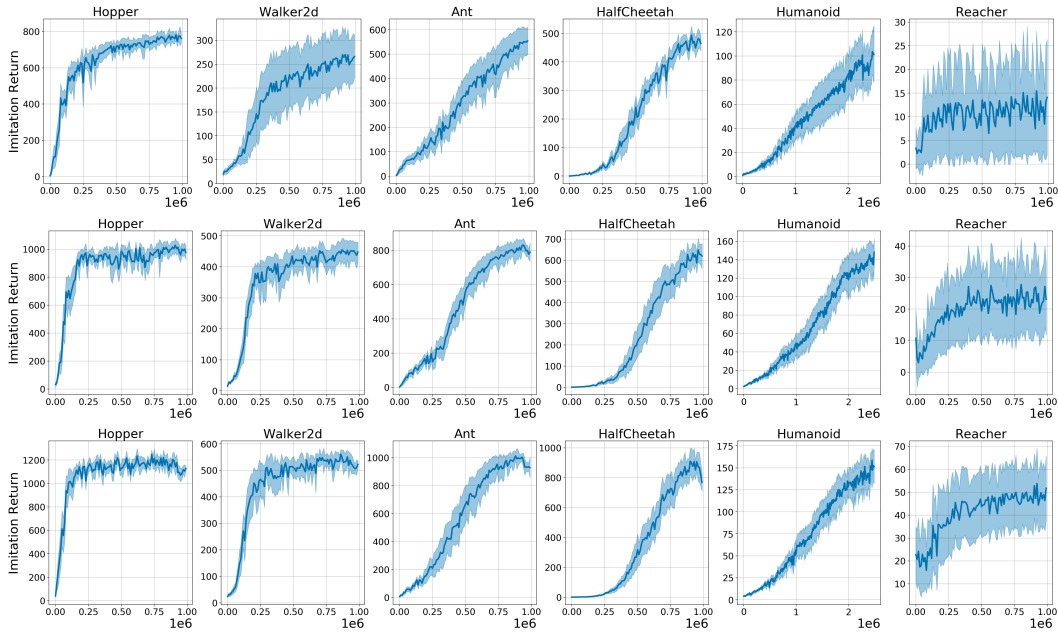

Figure 7: Mean and standard deviation of the imitation return of the agent. We perform 10 rollouts over 10 seeds every 10k environment steps over 1M steps. The return here is in term of the reward defined with Equation (6). Top row: 1 demonstration, middle row: 4 demonstrations, bottom row 11 demonstrations.

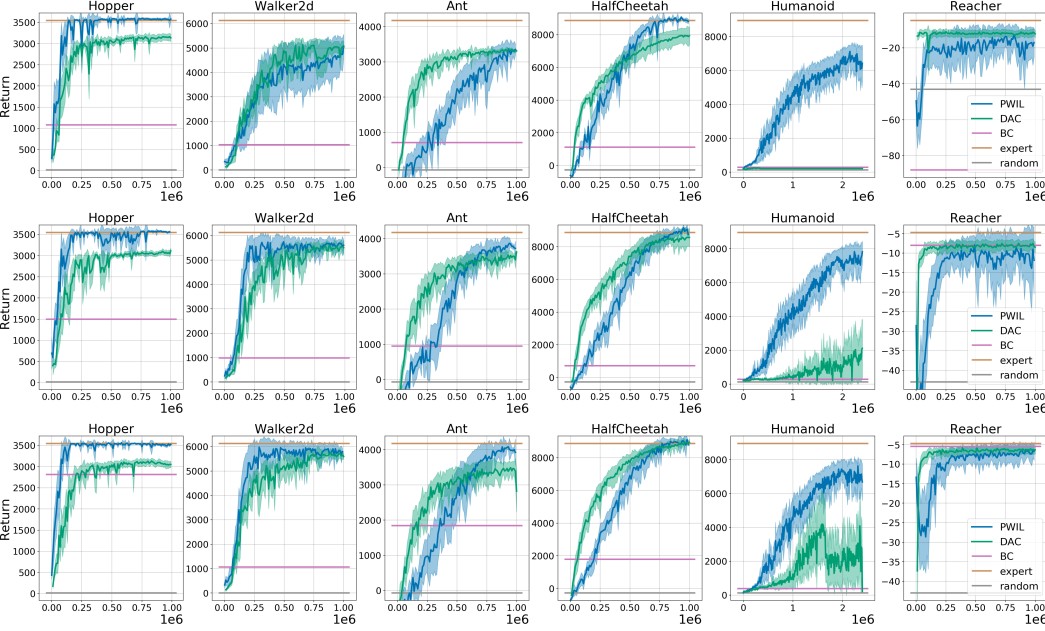

Figure 8: Mean and standard deviation of the original environment returns of the evaluation policy over 10 rollouts and 10 seeds, reported every 10k environment steps over 1M steps. Top row: 1 demonstation, middle row: 4 demonstrations, bottom row: 11 demonstrations.

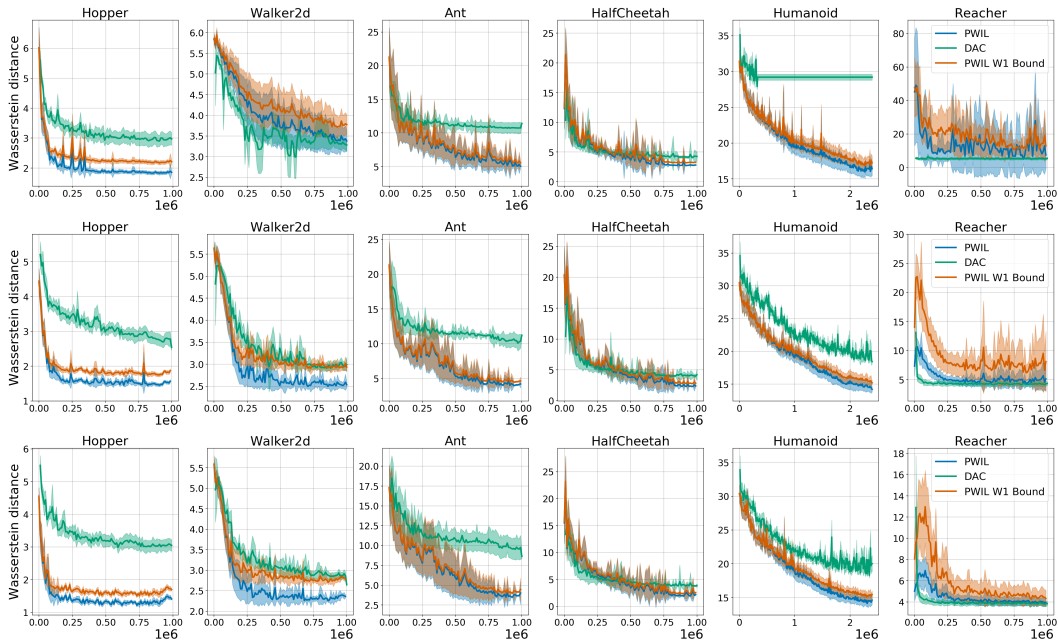

Figure 9: Mean and standard deviation of the Wasserstein distance between the state-action distribution of the evaluation policy and the state-action distribution of the expert over 10 rollouts and 10 seeds, reported every 10k environment steps. We include the upper bound on the Wasserstein distance based on the greedy coupling defined in Equation (4). Top row: 1 demos, middle row: 4 demos, bottom row: 11 demos.

# B  ABLATION STUDY

We present the learning curves of PWIL in the presence of ablations.

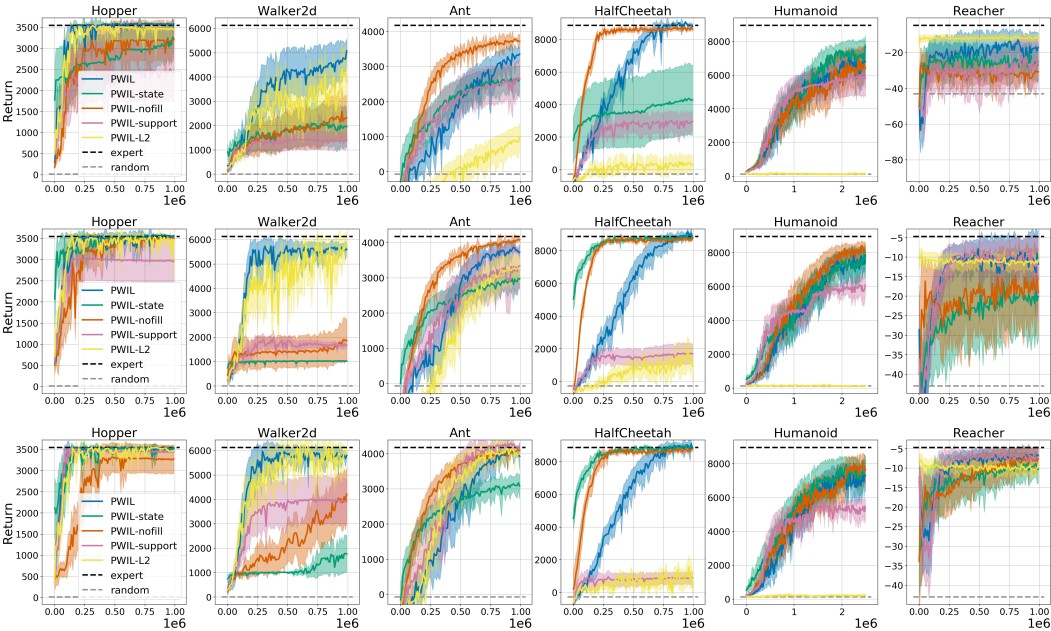

Figure 10: Evaluation performance of different variants of PWIL over 10 rollouts and 10 seeds, reported every 10k environment steps over 1M steps. The return is in term of the environment's original reward. Top row: 1 demonstration, middle row: 4 demonstrations, bottom row: 11 demonstrations.

## C   INFLUENCE OF THE DIRECT RL ALGORITHM

In this section, we study the influence of the direct RL method on the performance on PWIL on locomotion tasks. We compare the performance of PWIL with D4PG with the performance of PWIL with TD3 and SAC. We use a custom implementation of both SAC and TD3 which is a reproduction of the authors' implementation. For SAC we used the same default hyperparameters; for TD3 we removed the delayed policy update which led to better performance. We evaluate both methods on the same reward function as D4PG. We remove the pre-warming of the replay buffer for SAC. For TD3, we found out that we need different pre-warming for each environment to get good performance (contrary to D4PG): we used 1000 state-action pairs for Reacher, Hopper and Walker2d and no pre-warming for Humanoid, Ant and HalfCheetah.

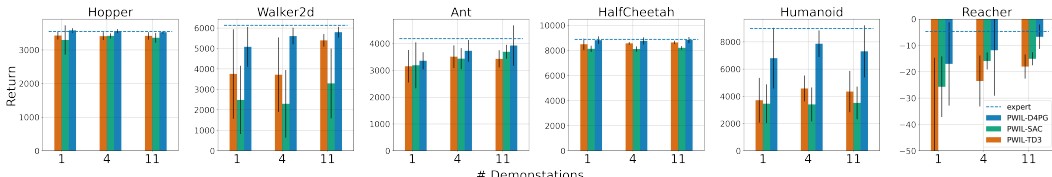

Figure 11: Mean and standard deviation of the evaluation performance of PWIL-D4PG, PWIL-SAC and PWIL-TD3 at the 1M environment interactions mark (2.5M for Humanoid). Results are computed over 10 seeds and 10 episodes for each seed.

We show in Figure 11 that both PWIL-SAC and PWIL-TD3 recover expert-like performance on Hopper, Ant and HalfCheetah and Reacher. For Walker2d, there is a large standard deviation in performance, since some seeds lead to near-optimal expert performance, and some seeds lead to poor performance. Remark that we led minimal hyperparameter tuning on the default versions of SAC and TD3, which might explain the differences in performance.

## D    DOOR OPENING EXPERIMENTS

In this section we present the details of the experiment presented in Section 4.4.

### D.1    ENVIRONMENT

The default door opening environment has a fixed horizon of 200 timesteps. We add an early termination condition to the environment when the door is opened to demonstrate that PWIL extends to tasks where there is no incentive for survival. Note that similarly to AIRL (Fu et al., 2018), early termination states are considered to be absorbing. Therefore if the expert reaches an early termination state early, its state distribution as a large weight on these states, which leads the PWIL agent to similarly reach the early termination state. 25 demonstrations were collected from humans using a virtual reality system (Kumar & Todorov (2015)).

### D.2    EMBEDDING

As we assume that we only have access to the visual rendering of the demonstrations (with resolution $84 \times 84$), we build a lower dimensional latent space. We use a self-supervised learning method: TCC, to learn the latent space. We used the classification version of the algorithm detailed in Dwibedi et al. (2019) (Section 3.2).

The encoding network is the following: a convolutional layer with 128 filters of size 8 and stride 4, a relu activation, a convolutional layer with 64 filters of size 4 and stride 2, a relu activation, a convolutional layer with 64 filters of size 3 and stride 1, a relu activation, a linear layer of size 128, a relu activation and an output layer of size 32.

We separated the demonstrations into a train set of 20 demonstrations and a validation set of 5 demonstrations. Similarly to Dwibedi et al. (2019), we select the encoding network by taking the one that maximizes the alignment according to Kendall's tau score on the validation set.

We ran an hyperparameter search over the batch size $\{1, 4, 16, 32\}$, the subsampling ratio $\{1, 4, 8\}$ and the learning rate $\{10^{-2}, 10^{-3}, 10^{-4}\}$ (using the Adam optimizer). The best performing encoder had an average Kendall's tau score of 0.98 on the training set and 0.91 on the validation set.

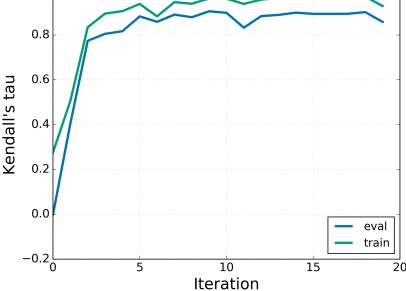

Figure 12: Kendall's tau score of the encoding network on the train and validation sets. An iteration corresponds to 1000 gradient updates.

### D.3    AGENT

In this setup, we found that SAC leads to better results than D4PG. We detail the implementation of SAC below.

The actor is a 3-layer network that outputs the parameters of a normal distribution with diagonal covariance matrix. We enforce the terms of the covariance matrix to be greater than $10^{-3}$. The first layer has size 256 with relu activation, the second layer has size 256 with relu activation and the last layer has size 48 (the action space has size 24).

The critic is a 3-layer network. We pass the concatenation of the state and the action of the environment as an input. There are two hidden layers of size 256 with relu activation. We do use a twin critic.

We use an adaptive temperature $\alpha$ as described by Haarnoja et al. (2018b).

For the three losses we use the Adam optimizer, and we use Polyak averaging for the critic networks with a rate $\tau$. We ran an hyperparameter search over the learning rates $\lambda_a, \lambda_c, \lambda_\alpha$ in $\{10^{-4}, 3 \times 10^{-4}\}$, over $\tau$ $\{0.01, 0.005\}$, over the batch size $\{64, 128, 256\}$ and over the number of interactions with the environment between each actor, critic and temperature update $\{1, 4, 8\}$.

## E  ALGORITHM DETAILS

### E.1  RUNDOWN

We present an example rundown of PWIL in Figure 13.

**Start episode.**

Figure 13: PWIL example rundown. We drop the dependency on the action. The example task has an horizon $T = 2$, and the demonstration set has $D = 4$ elements. We take $f : c \mapsto \exp(-c)$.

### E.2  RUNTIME

We detail the runtime of PWIL in this section. A single step reward computation consists in computing the distance from the current observation (a vector with dimension $|\mathcal{S}| + |\mathcal{A}|$ with $D$ vectors (the expert observations), which has complexity $\mathcal{O}((|\mathcal{S}| + |\mathcal{A}|)D)$. The next step is to get the minimum of these distances (complexity $\mathcal{O}(D)$), for $\lceil \frac{D}{T} \rceil$ times (the number of times the algorithm executes the while loop). Therefore the complexity of a single step reward computation is $\mathcal{O}((|\mathcal{S}| + |\mathcal{A}|)D + \frac{D^2}{T})$ and the complexity of the reward computation of an episode is $\mathcal{O}((|\mathcal{S}| + |\mathcal{A}|)DT + D^2)$.

