# OpenReview forum: "Primal Wasserstein Imitation Learning"
_ICLR.cc/2021/Conference — ICLR 2021 Poster_

### Official Review · AnonReviewer4 · 2020-10-28
**Wassertein Distance based immitation learning with greedy approximation learns from fewer exemplars.**

**Rating:** 6
**Confidence:** 3

**Review:**

The authors present a well written paper with many detailed experiments.  I am happy to revisit my evaluation after a fruitful discussion :)

Pros

Wesserstein distance between demonstration and policy is setup
Using the earth mover’s analogy with the knowledge of T piles of dirt, distances to the holes are computed. Then the piles of dirt are moved to the closest holes in a greedy fashion.
An offline learning policy is presented
Sample complexity for learning is suggested to be low
The model seems to be able to learn from a single example to run

Cons

The Sample complexity for training is still high. While I do agree that the number of demonstrations required is lower.

The win is not obvious or strong except for the humanoid condition. This is my biggest concern. Can we explore more complex environments? Or the effects of more demonstrations? It would be nice to understand where this model truly succeeds or fails.

In the door opening experiment it would be great to know how the model performs while varying the number of expert demonstrations (say).

Might be tangential, but this paper seems relevant at least in introducing the idea of inverse reinforcement learning and W-distance
https://arxiv.org/pdf/1906.08113.pdf

---

> ### Author Response · Authors · 2020-11-13
> **Response to Reviewer 4**
>
> We thank the reviewer for their feedback.
>
> *“The Sample complexity for training is still high. While I do agree that the number of demonstrations required is lower.”*
>
> We assume that the reviewer is referring to “sample complexity” as the number of interactions with the environment. We point out to the reviewer that the sample complexity of PWIL is comparable to DAC which is state-of-the-art in the MuJoCo domain. ValueDice has better sample efficiency but the setup is different since there is no subsampling, and in fact performs poorly in the subsampling setting (we can include plots if the reviewer deems necessary).
> For reference, on Humanoid, the only other imitation learning method that reaches non-trivial behavior (running vs standing) is GAIL (to the best of our knowledge): GAIL uses 75M environment interactions against 2.5M for PWIL, while also using significantly more expert trajectories (>80).
>
> *“The win is not obvious or strong except for the humanoid condition. This is my biggest concern. Can we explore more complex environments? Or the effects of more demonstrations? It would be nice to understand where this model truly succeeds or fails.”*
>
> While we agree with the reviewer that the performance gap is only significant on Humanoid (and we might add Hopper), we insist that Humanoid is the most complex MuJoCo locomotion task (again, only GAIL reaches non-trivial behavior but with considerably more demonstrations and environment interactions). We also point out that the number of hyperparameters used with PWIL is significantly lower than adversarial IL approaches which should be considered a “win”.
>
> We point out to the reviewer that the relevant related work rarely goes beyond MuJoCo locomotion tasks, and that we showed how PWIL can adapt to a harder setup in Section 4.4. The door experiment is not a locomotion task, and is complex since: 1) it is not a repetitive sequence of actions (e.g. moving one leg after the other), 2) the demonstrations are given in the form of pixels, rather than the inner state of the controller, 3) we do not have access to the actions.
>
> Finally, while we agree with the reviewer that we do not present an environment where PWIL fails, we believe that we did a considerable effort (which is usually absent in the relevant related work) in the ablation study to highlight how the different components of PWIL explain its success and failure modes. For example, we showed that PWIL is sensitive to the quality of the metric (PWIL-L2); we showed that PWIL goes beyond expert support estimation methods (PWIL-support); we showed that PWIL can work even in the absence of expert actions (PWIL-state).
>
> *"In the door opening experiment it would be great to know how the model performs while varying the number of expert demonstrations (say)."*
>
> We agree with the reviewer that it would be great to have. We will try to incorporate the results by the end of the discussion period, if we cannot make it on time, we will make sure to incorporate the results in the camera-ready version.
>
> *"Might be tangential, but this paper seems relevant at least in introducing the idea of inverse reinforcement learning and W-distance https://arxiv.org/pdf/1906.08113.pdf"*
>
> This paper was actually cited in our work. We discussed in the related work section how their approach is different as it is based on a dual formulation of the Wasserstein distance (which still involves a minmax optimization problem).

---

### Official Review · AnonReviewer1 · 2020-10-29
**Updated review for PWIL**

**Rating:** 6
**Confidence:** 4

**Review:**

The paper develops an imitation-learning (IL) method starting from the primal form of the Wasserstein distance, creating an upper-bound (by replacing optimal coupling with a greedy coupling), and converting that into a practical, scalable algorithm (PWIL). Experiments on the standard MuJoCo tasks and a pixel-based door opening task show that the method is competitive to the recent IL approaches in terms of performance and sample-efficiency.

I am currently on the fence about this paper. I would like the authors to address the following concerns:

1.	The experiments in Figure 2 compare PWIL (Distributional DDPG) with DAC (TD3). The difference in the underlying off-policy RL algorithm somewhat muddles the comparative evaluation of the reward-deriving strategy – greedy coupling in PWIL vs. discriminator training in DAC. From what I understand, PWIL should work seamlessly with TD3 as well. Is there a particular reason why the authors went with distributional-RL? Providing PWIL+TD3 results would be an equitable comparison to DAC.

2.	Discussion of potential limitations of the current approach -- the paper does strong relaxations when moving from theory to practice. I don’t have issues about this, but it would certainly be useful to include comments about the potential pitfalls of the approach. For instance, are there MDPs where the following the greedy coupling strategy provides deceptive rewards leading to sub-optimal imitation? Another example is the handling of the reward bias. Equation (6) is explicitly biased to provide a positive reward, and therefore, cannot handle cases where the episodes should be terminated at the goal (like the popular LunarLander Gym environment). Section 4.4 adds a manually designed constraint to get around this, but unlike DAC, I do not see a principled way to manage this with PWIL.

3.	Comparing rewards to those from adversarial IL – In the case of adversarial IL, the rewards are non-stationary as the discriminator parameters change throughout the training. However, for discriminator parameters at any particular iteration, the rewards are Markovian (depend only on current state and action). The rewards for PWIL seem to be non-Markovian (Equation 5), yet they have been integrated with the RL machinery (policy-gradients like DDPG) designed for Markovian rewards. Is this observation correct, and if yes, is there an intuition for why this should work?

4.	I assume the blue curve in Figure 3 is the actual Wasserstein distance. It would be helpful to add a line or two regarding the method used to obtain the optimal coupling to plot this.


===== POST REBUTTAL UPDATE =====


I have updated my score based on the response. I hope the authors can add PWIL+TD3 results (point 1. in review) in the next revision.

---

> ### Author Response · Authors · 2020-11-13
> **Response to Reviewer 1**
>
> We thank the reviewer for their feedback.
>
> 1. The reviewer is right that PWIL+TD3 would be an interesting comparison. The choice of D4PG was made as it is a reliable DDPG-like agent available on ACME. We will try to incorporate PWIL+TD3 by the end of the reviewing period (which might be a long shot) and we will make sure to include it in the camera-ready version.
>
> 2. The reviewer is very right that there are some limitations as we go from theory to practice.
> For the suboptimality of the greedy coupling, we provided an example in Figure 1 to give an intuition of how it can be suboptimal and we will emphasize on how this constitutes a limitation of our work.
> For the handling of the reward bias, we are not sure how our method is less principled than DAC. Conceptually, DAC adds an absorbing state (in the authors’ implementation a vector with zeros) to the MDP at hand to indicate a terminating state (which in fact is only well adapted to environments where terminating states are either all desirable or all undesirable). In Section 4.4, we also extend the MDP with self-transitions to terminating states rather than transitions from terminating states to an absorbing state. We will indicate the difference between our approach and DAC.
>
> 3. The reviewer is right in the fact that PWIL’s reward function is non-Markovian, and that for a fixed discriminator, the reward function of an adversarial IL approach is Markovian. However, the discriminator is continuously changing (within an episode) which makes the reward function of adversarial IL approaches effectively non-Markovian as well. We agree with the reviewer that exploring how these direct RL methods (TRPO for GAIL, TD3 for DAC, DDPG for PWIL) works with non-Markovian rewards is an interesting avenue of research.
>
> 4. We agree with the reviewer and explain how the optimal coupling is derived in the revised version.

---

### Official Review · AnonReviewer2 · 2020-10-29
**Review for Paper1564**

**Rating:** 8
**Confidence:** 5

**Review:**

### Summary
The authors proposed an imitation learning algorithm that utilizes the primal form of Wasserstein distance to match agent’s and expert’s state-action visitation distributions. They considered the upper bound of the primal form and devise the optimization method based on greedy coupling which makes learning suitable for sequential problems. With standardized Euclidean distance and exponential smoothing, the proposed method PWIL is shown to perform well for both MuJoCo and Door Opening Tasks and highly outperforms the baseline (DAC) for Humanoid.

### Quality
Theoretical derivations are sound, and experiments are well-tuned and designed and highly support the authors’ claims.

### Clarity
The submission is clearly written overall, but I added minor comments on clarity in `Detailed Comments`

### Originality
The idea of submission is novel from the use of the primal form of Wasserstein distance in concatenation with greedy coupling.

### Significance
I think the contribution of this work is significant in the sense that it considers the primal form of Wasserstein distance and eventually show the decrease of Wasserstein distance through experiments, whereas the existing works haven’t explicitly shown the performance in terms of probabilistic metrics.

### Detailed comments
(p. 1, Abstract) We present a reward function which is derived offline, as opposed to recent adversarial IL algorithms that learn a reward function through interactions with the environment, and which requires little fine-tuning.
- I agree that PWIL doesn’t require neural networks and relevant gradient updates for reward learning but wasn’t sure about the term “offline” is appropriate to describe this. PWIL involves the internal loop for reward updates during environment simulation. This is a bit confusing at the first glance since I believe “offline” is used when the environment interaction is not allowed, e.g., offline reinforcement learning. (My first bias was that the reward is learned before the interaction, fixed, and then used during training without any updates.) I guess there’ll be a better expression for it (like a non-parameter reward?) or simply ignoring the term “offline” is much better.
- A sentence in the introduction “The inferred reward function is non-stationary, like adversarial IL methods, but it is not re-evaluated as the agent interacts with the environment, therefore the reward function we define is computed offline.” tries to clarify this term, but I don’t understand why it is said that the reward is *not re-evaluated* since the most inner loop of **Algorithm 1** is related to the reevaluation of rewards.

(p. 1, Introduction) Our method recovers expert behaviour better than existing state-of-the-art methods while being based on significantly fewer hyperparameters
- In Figure 1, PWIL seems to perform comparably to DAC, not better than DAC, except for Humanoid.

(p. 2, Background and Notations) $\delta$ is the discount factor
- It seems like $\delta$ is used instead of $\gamma$ since $\gamma$ is used to indicate a stochastic matrix. However, since most potential readers may be familiar with the terms in RL and imitation learning, I’d rather use $\gamma$ for the discount factor and a different letter for coupling.
- Also, it collides with Drac distribution in the following paragraph.

(p. 2, Background and Notations) from maximizing to minimizing its return
- $\rightarrow$ from maximizing its return to minimizing cumulative costs?

(p. 2, Background and Notations) requires the definition of a metric in the space
- $\rightarrow$ requires the definition of a metric $d$ in the space (to easily link with one in the definition of Wasserstein distance)

(p. 2, Method) We introduce a reward based on an upper-bound of the Wasserstein distance
- $\rightarrow$ We introduce a reward based on an upper-bound of the Primal form of  Wasserstein distance

(p. 3, Wasserstein distance minimization) This can be problematic if an agent learns in an online manner or in large time-horizon tasks. Thus, we introduce an upper bound to the Wasserstein distance that yields a cost we can compute online, based on a suboptimal coupling strategy.
- I agree that this is a key idea of the submission, but I was wondering if it isn’t available to calculate the cost even when off-policy RL is applied as was done in the experiments.

(p. 3, Greedy Coupling) Eq (4)
- It seems to me removing $\inf_{\pi\in\Pi}$ is possible and fits the whole equation inside the format.

(p. 4, Figure 1) Note that the total cost with the greedy coupling is 7 whereas the total cost with the optimal coupling is 5.
- With the definition in Eq (5), I think the multiplication of $\gamma_\pi^q[i, j]$ (which is the uniform distribution) is missing.
- Also, it would be better to add detailed calculations in the Appendix for curious readers.

(p.4, Greedy Coupling)  The algorithm computes the greedy coupling with a complexity $\mathcal{O}((|\mathcal{S}|+|\mathcal{A}|)D)$
- How’s the complexity calculated in detail?

(P.4, Experiments) As DAC is based on TD3 which is a variant of DDPG, we use a DDPG-based agent for fair comparison.
- This makes me a bit confused since it was non-trivial for me to link off-policy RL with Algorithm 1 directly. One thing I’m really curious about is how the reward comes from the replay buffer used in a DDPG-based agent.

(P.4, Experiments) Figure 2
- How’s the computational cost for overall evaluation? Specifically, I think there’s no problem with the evaluation of Wasserstein distance when a small number of the expert demonstration is given, but the calculation time will hugely increase for a large number of expert data (which I believe can be in practical scenarios).

---

> ### Author Response · Authors · 2020-11-13
> **Response to Reviewer 2**
>
> We thank the reviewer for the detailed feedback and the strong recommendation, we will directly incorporate the minor comments in the revised version and address the questions below.
>
> *On PWIL’s reward function is “offline”.*
>
> We agree with the reviewer that “offline” is overloaded. The justification here is that the reward function r(s_t, a_t) defined by PWIL is a function f(s_0, a_0, …, s_t, a_t) that can be derived offline e.g. without interactions with the environment. The confusion probably comes from the fact that this function f is history dependent. We will clarify this in the updated version.
>
> *“The algorithm computes the greedy coupling with a complexity O((|S|+|A|)D)”*
>
> The reviewer is right, there is a mistake here. The complexity of the a single reward step computation is O((|S|+|A|)D + D^2/T). We provide details in the updated version.
>
> *“This makes me a bit confused since it was non-trivial for me to link off-policy RL with Algorithm 1 directly. One thing I’m really curious about is how the reward comes from the replay buffer used in a DDPG-based agent.”*
>
> We make it clearer in the revised version how Algorithm 1 relates to off-policy methods. The idea is that A.observes adds a (s, a, r) tuple to the replay buffer, and A.updates samples a batch and updates the policy and the critic. The reviewer makes an excellent point as a fundamental difference between DAC and PWIL is the following: the reward of PWIL is derived as the agent interacts with the environment and is added to the replay buffer. In DAC, only state and action pairs are added to the replay buffer and the reward is derived with the current version of the discriminator as the state-action batch is sampled.
>
> *“I agree that this is a key idea of the submission, but I was wondering if it isn’t available to calculate the cost even when off-policy RL is applied as was done in the experiments.”*
>
> If we understand correctly, the reviewer suggests whether we could use the optimal coupling rather than the greedy coupling. This is indeed an interesting direction for future work since it is possible for some direct RL agents to play a complete trajectory, compute the optimal coupling and compute rewards a posteriori.
>
> *How’s the computational cost for overall evaluation? Specifically, I think there’s no problem with the evaluation of Wasserstein distance when a small number of the expert demonstration is given, but the calculation time will hugely increase for a large number of expert data (which I believe can be in practical scenarios).*
>
> The computational cost is based on the default method from the POT library [1], which is in O(max(T, D)^3) (note that it’s considerably less expensive in practice [2]). The reviewer is right that this cost can become intractable as D becomes significantly larger. However, in the presence of a large number of demonstrations (which is not the setup here), we believe that BC is more competitive and alleviates the need for environment interactions.
>
> [1] Rémi Flamary and Nicolas Courty, POT Python Optimal Transport library,
> Website: https://pythonot.github.io/, 2017
>
> [2] Bonneel, N., Van De Panne, M., Paris, S., & Heidrich, W. (2011, December). Displacement nterpolation using Lagrangian mass transport. In ACM Transactions on Graphics (TOG) (Vol. 30, No. 6, p. 158). ACM.

---

### Official Review · AnonReviewer3 · 2020-10-30
**Review for paper "Primal Wasserstein Imitation Learning"**

**Rating:** 6
**Confidence:** 4

**Review:**

Summary:
This paper proposes to use Wasserstein distance in the primal form for imitation learning. Compared with its dual form and f-divergence minimization variants, it avoids the unstable minimax optimization. In order to compute the Wasserstein distance in primal form, they also propose a greedy approximation. Their experiments demonstrate that this method has a better performance compared with baseline methods.

Pros:
+ The greedy approximation of the primal Wasserstein distance is a clean solution, and it works well for MuJoCo tasks, even if in the LfO setting
+ The paper presents a series of experiments and ablation studies, which showed quite strong performance. In the ablation, the experiments about PWIL-support is quite convincing. The agent may stay on the supports for other the density/support estimation-based methods, while PWIL doesn't.
+ The paper is well-written and easy to follow. It provides enough implementation details and codes, which is reproducible

Some questions & concerns:
+ I'm wondering if the uniform distribution assumption can be improved, by incorporating certain density estimation methods.
+ Why the complexity of the algorithm is O((|S| + |A|)D)? It should also be linear in T
+ One limitation is that you need carefully selected metrics, e.g. L1, standardized L2. And it's generally hard to compute Wasserstein distance in the image domain.
+ For the visual imitation experiments, the figures only contain PWIL without the results of other baselines. They can still be trained on the feature from TCC.

---

> ### Author Response · Authors · 2020-11-13
> **Response to Reviewer 3**
>
> We thank the reviewer for their feedback.
>
> *“I'm wondering if the uniform distribution assumption can be improved, by incorporating certain density estimation methods.”*
>
> While we agree that density estimation methods for imitation learning are an interesting research direction, the motivation of PWIL is opposite. Density estimation methods are well adapted (and in fact sometimes necessary) for f-divergence based approaches. The problem is that estimating a density becomes hard when the dimensionality increases (e.g Section 4.4) or in the presence of very few demonstrations (e.g. MuJoCo experiments with 1 demonstration with a subsampling ratio of 20). The Wasserstein distance alleviates the need to estimate a density and computes a distance between distributions by using the geometry inferred from the distance in the MDP.
>
> *“Why the complexity of the algorithm is O((|S| + |A|)D)? It should also be linear in T”*
>
> The reviewer is right in pointing out that “The algorithm computes the greedy coupling with a complexity O((|S| + |A|)D).” is erroneous. The complexity for a single step reward computation is O((|S| + |A|)D + D^2/T) and it is O((|S| + |A|)DT +D^2) for the whole episode. The details of the derivations are given in the revised version.
>
> *“One limitation is that you need carefully selected metrics, e.g. L1, standardized L2.”*
>
> We agree with the reviewer that PWIL necessitates a metric on an MDP which is a strong requirement. However the point of Section 4.4 is precisely to demonstrate that even in cases where the metric is not obvious (pixel-based), we provide a method to learn it and show that PWIL can solve the task at hand. Furthermore, we actually highlighted in the ablation study that PWIL is sensitive to the quality of the metric used since it does not perform as well with the L2 distance as with the standardized L2 distance in the MuJoCo domains (see PWIL-L2). However, we do not think that these metrics are “carefully” selected: using a standardized L2 distance is actually quite natural since the observation dimensions are not on the same scale.
>
> *“And it's generally hard to compute Wasserstein distance in the image domain.”*
>
> We agree with the reviewer’s statement. It is “generally hard” to compute the Wasserstein distance between two distributions of images (using the L2 distance). However, in this case, we use the sequential aspect of the demonstrations to learn a metric using TCC, which facilitates the derivation of the Wasserstein distance. And, as we use the greedy coupling as an approximation to the Wasserstein distance, it becomes even easier to derive. The point of Section 4.4 is precisely to demonstrate that PWIL can extend to the image domain.
>
> *“For the visual imitation experiments, the figures only contain PWIL without the results of other baselines. They can still be trained on the feature from TCC.”*
>
> The baselines used in the MuJoCo experiments (BC and DAC) do not extend in the LfO setting. More suitable baselines here might be BCO or GAIfO. However, the point of Section 4.4 is to show that PWIL can extend to harder setups than MuJoCo locomotion tasks (LfO, reward defined from pixels), where the MDP metric has to be learned.

---

### Decision · Program_Chairs · 2021-01-07
**Final Decision**

**Decision:**

Accept (Poster)

**Comment:**

It is common in imitation learning to measure and minimize the differences between the agent’s and expert’s visitation distributions. This paper proposes using Wasserstein distance for this, named PWIL, by considering the upper bound of its primal form and taking it as the optimization objective. The effectiveness of the approach is demonstrated by an extensive set of experiments.

Overall, reviewers reached general agreement that this paper makes a good contribution to the conference, and given the overall positive reviews, I also recommend accepting the paper.